# Emerging Roles of Airway Epithelial Cells in Idiopathic Pulmonary Fibrosis

**DOI:** 10.3390/cells11061050

**Published:** 2022-03-19

**Authors:** Ashesh Chakraborty, Michal Mastalerz, Meshal Ansari, Herbert B. Schiller, Claudia A. Staab-Weijnitz

**Affiliations:** Member of the German Center for Lung Research (DZL), Institute of Lung Health and Immunity and Comprehensive Pneumology Center with the CPC-M BioArchive, Helmholtz Zentrum München GmbH, 81377 Munich, Germany; ashesh.chakraborty@helmholtz-muenchen.de (A.C.); michal.mastalerz@helmholtz-muenchen.de (M.M.); meshal.ansari@helmholtz-muenchen.de (M.A.); herbert.schiller@helmholtz-muenchen.de (H.B.S.)

**Keywords:** basal cells, bronchial epithelium, airway epithelium, lung fibrosis, MUC5B, single cell RNA sequencing, epithelial populations, IPF

## Abstract

Idiopathic pulmonary fibrosis (IPF) is a fatal disease with incompletely understood aetiology and limited treatment options. Traditionally, IPF was believed to be mainly caused by repetitive injuries to the alveolar epithelium. Several recent lines of evidence, however, suggest that IPF equally involves an aberrant airway epithelial response, which contributes significantly to disease development and progression. In this review, based on recent clinical, high-resolution imaging, genetic, and single-cell RNA sequencing data, we summarize alterations in airway structure, function, and cell type composition in IPF. We furthermore give a comprehensive overview on the genetic and mechanistic evidence pointing towards an essential role of airway epithelial cells in IPF pathogenesis and describe potentially implicated aberrant epithelial signalling pathways and regulation mechanisms in this context. The collected evidence argues for the investigation of possible therapeutic avenues targeting these processes, which thus represent important future directions of research.

## 1. Introduction: An Emerging Role of the Airway Epithelium in IPF Aetiology

Idiopathic pulmonary fibrosis (IPF) is characterized by excessive deposition of extracellular matrix (ECM) within the alveolar compartment of the lung, leading to impairment of gas exchange, increased stiffness and, ultimately, loss of lung function. Despite approval of the two first effective antifibrotic drugs more than six years ago [1,2] and intensive sustained efforts in clinical drug development, IPF remains associated with high mortality rates. Current therapeutic options do not halt disease progression and prevalence of IPF appears to be rising worldwide [3].

The aetiology of IPF is incompletely understood. Traditionally, IPF was believed to be mainly caused by repetitive injuries to the alveolar epithelium. A growing body of evidence, however, based on genome-wide association studies (GWAS), molecular profiling of patient samples, high-resolution micro-CT imaging, and single cell RNA-Sequencing (scRNA-Seq), suggests that IPF equally involves an aberrant response of the bronchial and bronchiolar epithelium, which contributes significantly to disease development and progression. In this review, we summarize known alterations in airway structure, function, and cell type composition in IPF. We furthermore give a comprehensive overview on the genetic and mechanistic evidence pointing towards an essential role of the airway epithelium in IPF pathogenesis. Potential mechanisms of aberrant airway epithelial regeneration and, finally, possible therapeutic avenues targeting these processes are discussed.

## 2. General Airway Structure

The lung is structurally and functionally categorized into two regions, the conducting zone and the respiratory zone. The conducting airways consist of the trachea, the bronchi, and the conducting bronchioles, whereas the respiratory zone contains the areas of gas exchange, the terminal (respiratory) bronchioles and the alveoli (Figure 1A). The conducting airways are lined with a pseudostratified epithelium composed primarily of basal, club, goblet and ciliated cells, which play an essential role in the first-line defence against inhaled toxins, particles, and pathogens. Structure and cell type composition of the conducting airway epithelium gradually changes with increasing airway generations from a pseudostratified appearance with mainly ciliated next to secretory and basal cells over a simple columnar to a simple cuboidal epithelium, which harbours fewer ciliated cells and more secretory cells, particularly club cells. In contrast, the alveolar epithelium in the respiratory airways is lined with alveolar type 1 (AT1) and type 2 (AT2) cells, which, together with endothelial cells below and the interjacent basement membrane, make up the blood-air barrier for O_2_/CO_2_ exchange [4,5] (Figure 1A). Basal cells are established as the main human progenitor cells for all cell types in the pseudostratified epithelium lining the conducting airways [6] while AT2 cells give rise to AT1 cells in the alveoli [7]. More recently, in the murine lung, the bronchoalveolar duct junction at the transition between bronchioles and alveoli has been described to harbour additional multipotent stem cells, the so-called bronchoalveolar stem cells (BASCs). These can give rise to club and ciliated cells on the one hand and AT1 and AT2 cells on the other hand, in particular in response to injury [8]. Whether such a population exists in the human lung, however, is unclear to date.

## 3. Changes in Airway Morphology in IPF

### 3.1. Airway Dilation

In recent years, multiple evidence has emerged that strongly argues for considerable changes in airway morphology and physiology in IPF, which contribute to disease progression. For instance, clinical CT findings in IPF patients as well as experimental micro-CT imaging of explanted IPF lungs demonstrate that proximal and distal airways are dilated [9,10,11,12], which may explain why FEV_1_/FVC ratios for IPF patients are higher than expected [13,14]. This is in agreement with aerosol-derived airway morphometry and capnographic measurements, which equally show increased airway volumes in IPF patients [15,16]. While changes in conducting airway volumes seem independent of disease severity [16], they appear to facilitate the distinction between stable and progressive disease, hence bear prognostic value [9]. The underlying mechanisms for airway dilation in IPF are not fully understood. Traditionally, traction bronchiectasis and bronchiolectasis, caused by increased collagen deposition and contraction of the peripheral fibrotic areas, have been thought to “pull open” the bronchi and bronchioles, respectively [17,18]. This concept is supported by the observation that the quantity of fibroblast foci correlates with traction bronchiectasis in high-resolution CT (HRCT) scans [19]. However, considering the comparatively distant location of fibrotic areas relative to the affected airways in IPF, and recent findings on emerging proliferative epithelial cell type populations in IPF (discussed below in Section 4), it has been suggested that the HRCT pattern of traction bronchiectasis in IPF is rather caused by bronchiolar proliferation than by mechanical traction alone [20].

### 3.2. Increased Airway Wall Thickness (AWT)

Recent studies report increases in airway wall thickness (AWT). Verleden et al. performed clinical CT and micro-CT of IPF explant and donor lungs, in combination with matched histological examinations. The authors observed that, due to increased AWT, more small airways are visible in CT scans of IPF specimens [21], a finding which was very recently confirmed by Ikezoe et al. [12] (Figure 2A). Additionally, a retrospective analysis of clinical chest CT images by Miller et al. suggested that lungs of IPF patients display significant increases in AWT, notably already in early disease stages [22]. Here, the authors performed so-called Pi10 measurements, which rely on a series of experimental determinations of total airway and luminal airway areas at different luminal perimeters. For each patient, the airway wall areas are calculated by subtraction of the luminal airway from the total airway area and the square root of these values is plotted against the perimeter. Regression analysis allows for the determination of the airway wall thickness of a hypothetical airway with an internal perimeter of 10 mm, the Pi10, a measure, which can then be directly compared between patients and disease cohorts. Interestingly, and explicitly mentioned by the authors as a limitation of their study, the way Pi10 is determined implies that changes in the internal luminal area, e.g., altered mucus layers, may have impacted the findings. As altered mucociliary clearance and increased MUC5B expression indeed are important features of IPF airways (discussed in Section 5), this raises the question whether Pi10 measurements are affected by increased levels of airway MUC5B, for example.

### 3.3. Bronchiolar Abnormalities

Bronchiolar lesions involving abnormal bronchiolar proliferation and migration are typical features of IPF and represent regions of injury and active regeneration [23,24,25]. While the observed increase in bronchiolar proliferation has been interpreted to result in an increased number of bronchioles in IPF [14,23], recent evidence based on micro-CT imaging and histology suggests it more likely leads to dilation and distortion of the small airways [10,11,12,21] (Figure 2A). In contrast, the number of terminal bronchioles is even reduced in IPF [10,12,21]. Importantly, the latter observation was made in areas of mild fibrosis and the number of terminal bronchioles did not further decline in areas with more severe fibrosis, indicating that loss of terminal bronchioles is an early event in IPF [21]. In addition, it was demonstrated in two very recent independent studies that loss of terminal bronchioles correlates with honeycomb formation and that conducting airways directly lead into honeycomb cysts [10,12]. In agreement, early studies have demonstrated that peripheral cystic air spaces are ventilated, but represent physiological dead-space because they are not perfused [26]. This supports the concept that small airways are the origin of honeycomb cysts, abnormal peripheral airway spaces that will be discussed in more detail in the following.

### 3.4. Honeycomb Formation and Bronchiolization

In thoracic radiology, the term “honeycombing” refers to clustered cystic airspaces which typically are located in the subpleural region of the lung [27]. While clinical HRCT only detects honeycomb cysts with a diameter of about 1 mm and bigger, smaller honeycomb cysts are usually observed in histology [28]. Typical microscopic honeycomb cysts in IPF are small, subpleural, and localized in vicinity to fibrotic areas. Figure 2B (lower row, IPF) shows a collapsed honeycomb cyst characterized by KRT5^+^ KRT14^+^ CC10^−^ cells in close proximity to fibroblast foci. On a cellular level, these honeycomb cysts are characterized by p63^+^ KRT5^+^ airway epithelial-like cell types replacing the normal alveolar epithelium, a process termed bronchiolization [29]. Some honeycomb cysts appear to be composed of stratified layers of hyperplastic p63^+^ KRT5^+^ KRT14^+^ cells [25] (e.g., Figure 2B), while others display a pseudostratified mucociliary epithelium, containing ciliated, p63^+^ KRT5^+^ basal, and goblet cells expressing *MUC5B* as the main mucin component [25,30,31]. Whether honeycomb cysts derive from the small airways or from the alveolar epithelium as a result of ectopic bronchiolar differentiation is still controversially discussed. Considering the current knowledge about epithelial progenitor cells in the distal lung, bronchiolization could be a result of AT2 cells committing to an aberrant differentiation program [7], or derive from migrating basal cells [6] or BASCs [8,32] originating from the small airway or of bronchoalveolar duct junction, respectively. BASCs, at least in the mouse, can give rise to AT2 and club cells upon injury [32], but there is, to the best of our knowledge, no evidence that they can give rise to p63^+^ KRT5^+^ basal cell-like populations, which most frequently line bronchiolized areas in the IPF lung [23,25,30,33]. This, in contrast, has been unambiguously demonstrated for airway stem cells in distal lung regeneration after injury: After influenza infection of mice, for example, p63^+^ cells emerge in the bronchioles and form extra-bronchiolar parenchymal clusters of p63^+^ Krt5^+^ basal cells, despite of little *TP63*-expression in normal murine bronchioles [34,35]. Lineage tracing experiments performed in independent laboratories have demonstrated that these cells derive from a rare population of SOX2^+^ p63^+^ Krt5^+/−^ progenitor cells, but not from alveolar epithelial cells or BASCs [34,36,37]. Hence, studies in mouse models of lung injury have argued against an alveolar origin of bronchiolized areas in IPF and rather suggested that bronchiolization may originate from the airways.

However, it is important to mention in this context, that studies in human organoid culture systems have provided compelling evidence that, in contrast to mouse AT2 cells, human AT2 cells can give rise to Krt5^+^ basal cells. This differentiation capacity into KRT5^+^ basal-like cells was strictly dependent on adult human lung mesenchymal cells (AHLM) as feeder cells. The resulting Krt5^+^ basal cells expressed canonical basal cell markers (*SOX2*, *TP63*) in addition to genes typically associated with aberrant basal epithelial populations in IPF [38]. Interestingly. scRNA-Seq analysis of AHLM further revealed that during organoid culture mesenchymal subpopulations emerge that resemble such enriched in IPF lung tissue [38]. Collectively, these findings indicate that pathological mesenchymal cells in IPF generate a niche that is supportive of aberrant differentiation of human AT2 cells into KRT5^+^ basal cells. Whether this is what happens in IPF, too, remains elusive, but it is plausible that aberrant basal cells in IPF derive from both airway and alveolar epithelial cells.

In summary, airways are drastically altered in IPF, with changes that (1) include macroscopic morphological changes visible by clinical and experimental CT imaging (airway dilation, increased airway wall thickness, honeycomb cysts), (2) manifest in physiological parameters like increased dead-space ventilation and higher FEV_1_/FVC ratios, and (3) involve repopulation of the injured alveolar region with basal-like epithelial cells, which may be both airway- and alveolar-derived (Figure 1B and Figure 2). On a cellular level, recent scRNA-Seq analyses of IPF lungs have provided even more weight to the importance of airway-like cells in IPF and will be discussed in the following chapter.

## 4. Recent Insights from Single Cell RNA-Sequencing (scRNA-Seq) Studies

Since the advent of single-cell RNA sequencing (scRNA-Seq), several studies in the past five years have revolutionized the concept of epithelial cell populations in IPF. In the earliest study, Xu et al. isolated Epcam^+^/HTII-280^+^ cells from peripheral regions of control and IPF lung and subjected that cell population to scRNA-Seq. Initially, they found that the yield of Epcam^+^/HTII-280^+^ cells, classically reflecting AT2 cells, drastically decreased in IPF lungs. However, more interestingly, in IPF, Epcam^+^/HTII-280^+^ subpopulations emerged which expressed transcripts typically associated with conducting airways and extracellular matrix-expressing cells, at the expense of genes typically associated with AT2 function [39]. Overall, the authors identified four subpopulations of Epcam^+^/HTII-280^+^ cells in IPF including (1) normal AT2 cells, (2) cells which expressed Goblet cell-specific markers, (3) cells which expressed basal cell-specific markers, and (4) indeterminate cells, which expressed multi-lineage markers including such for AT2, AT1, conducting airway cells and mesenchymal cells, and could thus not unambiguously be assigned to one cell type. Remarkably, the latter often co-expressed *SOX2* and *SOX9*, genes that typically define proximal airway progenitor and distal airway progenitor cells in the adult lung, respectively, thus indicating a loss of proximal-distal patterning in the IPF lung. Notably, SOX2^+^/SOX9^+^ progenitor cells otherwise only emerge in human lung development during the pseudoglandular stage in the distal epithelium but are already lost in the canalicular stage [40]. In addition, a more recent study suggests that surfactant processing is lost in these newly emerging epithelial cell populations, adding an important functional outcome of these changes [41]. Hence, in summary, in IPF a drastic loss of normal AT2 cells is paralleled by an increase of conducting airway characteristics in peripheral alveolar epithelial cells and an activation of aberrant differentiation programs or possibly reactivation of early lung developmental programs.

While the study above analyzed sorted Epcam^+^/HTII-280^+^ cells, isolated from a limited number of control and IPF lungs (*n* = 3), four more recent studies analyzed single cell suspensions from more specimens, without prior experimental enrichment for epithelial cells [42,43,44,45]. For visualization of the most important and consistent findings regarding epithelial cell populations in IPF/interstitial lung disease (ILD), we generated an integrative data set comprising all four studies (Figure 3A–C) using the Scanpy package (v1.8.0) [46]. To address potential batch effects, the integration was performed as described in Mayr et al. [43]. Briefly, the publicly available raw count matrices were re-processed data set wise with the same procedure. To mitigate effects of background mRNA contamination, the matrices were corrected by using the function adjustCounts() from the R library SoupX [47]. The expression matrices were normalized with scran’s size factor based approach [48], log transformed via scanpy’s pp.log1p() and finally scaled to unit variance and zero mean before concatenating them. A shared set of variable genes was selected by calculating gene variability patient-wise (flavor = “cell_ranger”, n_top_genes = 4000) and excluding known cell cycle genes. The intersection of the variable genes across all data cohorts was used as input for principal component analysis (1311 genes). After subsetting to the epithelial cell populations, the BBKNN method [49] was used to generate a batch balanced data manifold (Munich: ILD = 7, controls *n* = 12; Chicago: ILD *n* = 9, controls *n* = 8; Nashville: ILD *n* = 20, controls *n* = 10; and New Haven: ILD = 32, controls *n* = 22). Cell type identities from the original publication were retained and harmonized across studies. All four studies consistently confirmed the concept of an emerging diverse repertoire of epithelial cell types in ILD including IPF, most strikingly an increase in cells with features of conducting airways at the expense of classical alveolar epithelial cells (Figure 3D).

In more detail, up to 10 distinct clusters of epithelial cells were defined in these studies. While all identified most classical epithelial cell types, i.e., AT1, AT2, basal, ciliated, and secretory cells by similar expression signatures (Figure 4), there are some differences in subcategorization of the described cell type clusters. For instance, while Habermann et al. [44] distinguished between ciliated cells and differentiating ciliated cells, such a distinction was not made in the other studies [42,45]. Furthermore, categorization of secretory cells differs significantly between these reports. Reyfman et al. categorized club cells based on *SCGB1A1* (also termed *CC10* or *CCSP*) expression and did not report goblet cells but *MUC5B*-expressing cells within their cluster of club cells [42]. Adams et al. distinguished between club and goblet cells, but in their report *SCGB1A1* expression is a characteristic of both cell types and club and goblet cells are differentiated from each other by *SCGB3A2* and *MUC5B* expression, respectively [45]. Published and unpublished results from our lab have shown that *SCGB1A1* is expressed by a subpopulation of MUC5AC^+^ goblet cells, too [50]; so indeed, *SCGB1A1* should rather be considered a more general marker for secretory cells than specifically for club cells. Possibly reflecting similar considerations, Habermann et al. refrained from the attempt to distinguish between club and goblet cells and instead defined several secretory cell type clusters based on expression of *SCGB1A1, SCGB3A2,* and *MUC5B* and combinations thereof. Collectively, these studies show that, at least based on single cell transcript analysis, there is a continuum of secretory cells with overlapping gene expression patterns, which are not easily sorted into club and goblet cells without information on cell shape, spatial distribution within the bronchial tree, and protein expression patterns. Therefore, here, we also refer to those as secretory cells, without further distinction into goblet and club cells (Figure 4 and Figure 5). Independent of secretory cell subcategorization, all studies consistently demonstrate an increase in secretory cells including MUC5B^+^ cells. This was equally observed in an independent scRNA-Seq study where the authors refer to SCGBB1A1^+^ MUC5B^+^ cells as club cells, which, as explained above, may not be entirely accurate due to the ambiguity of SCGBB1A1 as a marker in that context. Still, also this study clearly demonstrates an increase of secretory cells in IPF relative to the healthy lung [51]. Furthermore, beyond quantitative alterations in epithelial cell populations, all IPF/ILD airway subpopulations displayed many significantly upregulated genes in their expression signatures when compared to their healthy counterparts (Figure 5B).

Basal cells appear to be particularly important in the context of IPF aetiology and progression for several reasons. For instance, a basal cell signature detected in the bronchioalveolar lavage transcriptome in IPF patients was predictive of mortality, strongly suggesting that basal cells play a central role in IPF progression [31]. Basal cell numbers are drastically increased in ILD (Figure 3D) and novel basal cell subpopulations and characteristics have already been demonstrated before the scRNA-Seq era. In 2015, Jonsdottir et al. reported that p63^+^ KRT14^+^ cells overlay fibroblastic foci in IPF (see also Figure 2B) and displayed characteristics of epithelial-to-mesenchymal transition (EMT) [52]. Shortly after, using immunofluorescence studies, Smirnova et al. quantified KRT5^+^ and KRT14^+^ basal cell population in healthy and IPF lungs and equally observed a drastic increase of basal cell populations in the distal IPF lung and proposed KRT14^+^ as a marker for an aberrantly differentiating progenitor cell pool [25]. The above-mentioned scRNA-Seq studies confirm these findings, showing that *KRT14* is overexpressed in basal cells in ILD, and also a marker of aberrant basaloid cells, which will be described below [42,44,45].

A recent scRNA-Seq study focussed on changes in basal cell plasticity in IPF and defined basal cell heterogeneity in the normal and IPF lung in greater detail [53]. According to this study, basal cells in the healthy lung can be subdivided in at least four subpopulations, namely classical multipotent basal cells (MPB), proliferating basal cells (PB), secretory-primed basal cells (SPB), and activated basal cells (AB). Based on scRNA-Seq data, surface marker screening, as well as bronchosphere assays, the authors established CD66 as a surface marker for SPBs and demonstrated an increase of CD66^+^ KRT5^+^ SPBs in IPF. With the importance of MUC5B and thus secretory airway cells in disease aetiology, these observations put forward modulation of basal cell priming as a novel therapeutic strategy in IPF [53].

Interestingly, Habermann et al. as well as Adams et al. identified a novel epithelial cell population with features of basal cells, which exclusively emerged in pulmonary fibrosis, namely KRT5^−^/KRT17^+^ epithelial cells [44], or aberrant basaloid cells [45]. These cells are comparably rare (Figure 3D) and characterized by expression of basal cell markers like *TP63*, *KRT17*, *LAMB3*, and *LAMC2* (but not *KRT5*, see Figure 4), in combination with mesenchymal markers like *COL1A1*, *VIM, TNC*, and *FN1,* and markers of senescence like *CDKN1A* (Figure 5A) [44,45]. Expression of *SOX9* and other markers of a distal differentiation program suggested that these cells also display characteristics of alveolar epithelial cells. Furthermore, these cells also showed the highest expression levels of *MMP7*, encoding matrix metallopeptidase 7, the probably best-validated peripheral blood biomarker for IPF (Figure 5A). Using RNA in situ hybridization, KRT17^+^/COL1A1^+^ basaloid cells were shown to cover fibrotic foci in IPF lungs but were not detected in non-fibrotic controls [44]. Given that these cells display characteristics of conducting and respiratory airways, the cellular origin is not clear. ScRNA-Seq-based pseudo-time analysis has raised the possibility that both transitional AT2 and *SCGB3A2*-expressing secretory cells may act as precursors for aberrant basaloid cells [43,44], a hypothesis which still requires experimental validation. Notably, studies in mouse models of lung fibrosis and injury have identified similar converging differentiation pathways, namely from club cells on the one hand and AT2 cells on the other to a population called Krt8^+^ alveolar differentiation intermediate (ADI) cells. This cell population is highly similar to the aberrant basaloid cells in IPF [54], but of transient character in bleomycin-induced lung fibrosis: Krt8^+^ ADI cells peak in the fibrotic phase and gradually disappear during resolution of fibrosis. Importantly, lineage tracing using Sox2- and Sftpc-Cre drivers has confirmed the dual, conducting airway and alveolar, origin of Krt8^+^ ADI cells. Collectively, this supports a model where an intermediate cell type, transiently emerging during a normal repair process, accumulates and persists in IPF.

In summary, scRNA-Seq studies have consistently demonstrated drastic changes in epithelial subpopulations in ILD, which strongly argue for an essential role of airway epithelial cells in disease development and progression. These include: (1) A dramatic decrease of normal alveolar cell types of the respiratory zone and their replacement by diverse conducting airway cell populations (Figure 3D). (2) The emergence of a novel ILD-specific cell type reminiscent of an intermediate cell involved in normal alveolar repair, which probably derives from both proximal and distal precursors and persists in lung fibrosis (Figure 3D and Figure 4). (3) Considerable changes in overall gene expression patterns in epithelial cell types (Figure 5).

## 5. Changes in Airway Function

### 5.1. Mucociliary Clearance

The discovery of the *MUC5B* polymorphism (see below, Section 6) has drawn a lot of attention to dysregulated mucociliary clearance as a major aetiological mechanism in IPF [29]. IPF is characterized by increased expression of *MUC5B* in the distal airways and honeycomb cysts. Increased expression is often driven by the minor allele (T) of the risk single nucleotide polymorphism (SNP) rs35705950, which is overrepresented in IPF patients. Consequently, the mucin MUC5B accumulates in airways of the distal lung where even mucous plugs can be observed within microscopic honeycomb cysts [55]. From other lung diseases, most prominently cystic fibrosis, it is very well known that overproduction of mucus impairs mucociliary clearance, leads to accumulation of particles and pathogens in the airways and increases the risk for chronic injury and inflammation. Indicating that this likely applies to lung fibrosis as well, *MUC5B* overexpression in distal airways has been shown to significantly impair mucociliary clearance and aggravate lung fibrosis in the mouse model of bleomycin-induced lung injury [56]. Importantly, in the same model, mucolytic treatment led to clearance of inflammatory cells from the lungs and counteracted the production of fibrillar collagen, providing proof-of-concept that restoring impaired mucociliary clearance may be beneficial in prevention and treatment of pulmonary fibrosis [56].

A potential key role of impaired mucociliary clearance for lung fibrogenesis is further emphasized by an independent study, where the issue of mucociliary clearance was approached from a very different angle. The E3 ubiquitin-protein ligase NEDD4-2 targets the epithelial Na^+^ channel (ENaC, encoded by *SCNN1A*) for intracellular degradation and thus plays a key role in limiting the levels of active ENaC at the cell surface. ENaC in turn is a critical regulator of epithelial surface hydration and consequently affects mucus properties. Overexpression of *SCNN1A* and activation of ENaC increases transepithelial transport of salt and water leading to dehydration of the apical epithelial mucous layer and thus impaired mucociliary clearance [57]. NEDD4-2 levels are decreased in IPF airways. With NEDD4-2 representing an antagonist of ENaC, conditional deletion of NEDD4-2 from airway epithelial cells in mice, as expected, increased ENaC activity and significantly impaired mucociliary clearance. A striking long-term consequence of this NEDD4-2 deficiency in murine airways, however, was the development of patchy lung fibrosis, bronchiolar remodelling, and increased MUC5B production in the peripheral airways, all features strongly reminiscent of IPF and actually reflecting IPF pathology more accurately than the most commonly used bleomycin-induced mouse model of lung fibrosis [58]. Collectively, these findings strongly indicate that mucociliary dysfunction is a major aetiological factor in IPF and, even though the minor risk allele within the *MUC5B* promoter will probably remain the most important cause, may have multiple origins including, e.g., dysregulation of epithelial surface hydration properties by NEDD4-2/ENaC.

### 5.2. Epithelial Barrier Dysfunction in IPF Pathogenesis

The bronchial epithelial barrier plays an important role in protecting the airways against environmental insults not only via mucociliary clearance and production of antimicrobial substances to eliminate inhaled pathogens, but also by tight junctions that maintain the cell–cell contact and regulate paracellular permeability [59]. Even if this has not been comprehensively assessed, some reports suggest that epithelial barrier function is altered during IPF pathogenesis. Zou et al., for instance, have demonstrated by immunohistochemistry (IHC) stainings for several tight junction proteins, that specifically levels of claudin-2 were elevated in IPF bronchiolar regions [60]. Others have found that levels of protein kinase D (PKD), a negative regulator of airway barrier integrity [61], were increased in IPF bronchiolar epithelium relative to normal lung tissue sections [62].

### 5.3. Other Changes in Airway Function

In a study designed to investigate the pathogenesis of cough in IPF, authors found increased levels of nerve growth factor and brain-derived neurotrophic factor in induced sputa of IPF patients compared to healthy control subjects [63]. These results indicated functional upregulation of sensory neurons in the proximal airways of IPF lungs.

## 6. Genetic Evidence Indicating Involvement of Bronchial Epithelium in IPF

IPF is a multifactorial disease where the interplay between environmental exposure and genetic susceptibility plays a central role in disease pathogenesis. Genome-wide association studies (GWAS) on large cohorts of various ethnical backgrounds have provided interesting insights into genetic susceptibility for IPF development and have linked specific genetic variants to poorer outcomes in sporadic IPF and familial pulmonary fibrosis [64]. In this context, single nucleotide polymorphisms (SNPs) conferring a higher risk for IPF were discovered in several genes reported to be expressed in airway epithelial cells, strongly suggesting a role for bronchial and bronchiolar epithelial cells in IPF aetiology [65,66]. These include mucin-5B (*MUC5B*), toll interactive protein (*TOLLIP*), desmoplakin (*DSP*), family with sequence similarity 13 member A (*FAM13A*), and A kinase anchor protein 13 (*AKAP13*). For all but *TOLLIP*, which seems comparably little expressed in airway epithelial cells, scRNA-Seq data confirms variable expression of these genes in bronchial, bronchiolar, and aberrant basaloid cells (Figure 6). While *MUC5B* and *FAM13A* are particularly expressed by secretory cells and ciliated cells, respectively, *DSP* is expressed by all bronchial and bronchiolar epithelial cell types including aberrant basaloid cells, where it is one of the top overexpressed genes relative to all other healthy epithelial cell types (Figure 5A). In contrast, except for *AKAP13*, expression of which is enriched in AT2 and aberrant basaloid cells, alveolar epithelial cells show relatively little expression of these genes (Figure 6).

### 6.1. MUC5B

A common promoter SNP in the airway gene *MUC5B* on chromosome 11, rs35705950, is the strongest risk factor for IPF, accounting for 30–35% of the overall risk to develop IPF [29,55]. *MUC5B* encodes mucin-5B, a mucin protein predominantly expressed in serous cells of submucosal glands in healthy lungs, and normally little expressed in airway surface epithelium [67]. In contrast, in IPF lungs, *MUC5B* is overexpressed in secretory cells within honeycomb cysts as well as in bronchioalveolar regions [29,30,39]. A series of elegant in vivo work has demonstrated that overexpression of *MUC5B*, both in proximal and distal airways, aggravates bleomycin-induced lung fibrosis in mice, while MUC5B-deficient mice are protected from the development of lung fibrosis. Interestingly, increased mortality was particularly observed when *MUC5B* was overexpressed in the distal murine airways [56].

### 6.2. TOLLIP

The gene *TOLLIP* encodes a ubiquitous protein with essential functions in the innate immune response, epithelial survival, defence against pathogens and further biological processes [68,69]. The *TOLLIP* gene is located adjacent to *MUC5B* and evidence regarding linkage disequilibrium between the *MUC5B* SNP rs35705950 and *TOLLIP* SNPs suggests that *TOLLIP* and *MUC5B* SNPs may not be passed on independently [68]. Three common variants within the *TOLLIP* locus (rs111521887, rs5743894, rs574389) have been shown to associate with higher susceptibility for IPF [66]. The minor alleles for all *TOLLIP* SNPs result in reduced expression by 20–50%, with rs111521887 and rs5743894, which are in high linkage disequilibrium, having stronger effects on expression than rs5743890 [66]. Interestingly, even though all result in reduced expression, the clinical effects of the rs111521887 and rs5743894 minor alleles are opposite to the rs5743890 minor allele: Individuals who carry the minor allele for rs111521887 and rs5743894 are more susceptible to developing IPF, while the minor allele rs5743890 is associated with less susceptibility. However, despite this initial protective effect, mortality in IPF patients with this variant is actually increased [66,68]. In the integrative scRNA-Seq data set that we examined, *TOLLIP* overall was comparably little detected (Figure 6). However, a recent study focusing on *TOLLIP* expression in the lung has demonstrated *TOLLIP* expression in AT2 cells, basal cells, and aberrant basaloid cells, but at the same time reported a global downregulation of *TOLLIP* expression in the IPF lung [70].

### 6.3. DSP

Linking intermediate filaments to the plasma membrane, desmoplakin, encoded by *DSP*, is a critical intracellular component of desmosomes, cell–cell adhesive junctions, which are critical for tissue integrity [71]. In the lung, *DSP* is primarily expressed in bronchi and bronchioles, with comparably little expression in alveoli [72]. The latter is also reflected by the scRNA-Seq data shown here (Figure 6). GWAS have linked at least two genetic variations in *DSP* with risk for IPF development, namely the minor alleles of rs2076295 and rs2744371 [65,72]. Among those, the minor allele of the intronic SNP rs2076295 (intron 5) is established as the strongest causal factor and is associated with an increased risk for IPF development, while the minor allele of rs2744371 confers a protective effect against IPF onset. Paradoxically, while *DSP* expression is increased in IPF lungs, the risk allele rs2076295 correlates with lower *DSP* expression. Some well-designed in vitro experiments using CRISPR/Cas9 gene editing in human bronchial epithelial cells have shown that deletion or disruption of the DNA region spanning rs2076295 as well as introduction of the minor allele (G) led to decreased expression of *DSP*, in agreement with an enhancer function of this region in intron 5 [73]. Decreased *DSP* expression in turn resulted in reduced barrier integrity, enhanced cell migration, and increased expression of markers for EMT and of ECM genes [73].

### 6.4. FAM13A

*FAM13A* encodes a so far uncharacterized protein with largely unknown function. Amino acid sequence homology suggests that FAM13A contains a Ras homologous (Rho) GTPase-activating protein (GAP) domain and hence a function in Rho GTPase signalling [74]. In the lung, *FAM13A* is primarily expressed in bronchial epithelial cells, but also by AT2 cells, and macrophages [75,76]. GWAS have identified a genetic risk variant within this gene, intronic rs2609255, that increases susceptibility for COPD and IPF with opposite risk alleles [65,77]. For IPF, this risk variant appears not to be associated with expression changes on transcript level [65]. Owing to its association with COPD and IPF disease risk, experimental studies have been performed in both disease contexts. These studies suggest that, on the one hand, FAM13A, protein levels of which are increased in COPD, may protect from cigarette smoke-induced disruption of airway integrity and neutrophilia [75], but at the same time promote β-catenin degradation, thus inhibit β-catenin signalling and associated repair processes, and increase susceptibility to emphysema [76]. On the other hand, FAM13A deficiency has been reported to exacerbate bleomycin-induced lung fibrosis in the mouse, possibly via induction of EMT-related gene expression [78]. Overall, FAM13A, even though its exact function remains unclear, appears to play an important role in airway epithelial barrier integrity and repair.

### 6.5. AKAP13

*AKAP13*, encoding A kinase anchor protein 13, is another gene with a genetic variant, rs62025270, conferring increased risk for development of IPF [79], expression of which is largely confined to the airway epithelium [80]. *AKAP13* is overexpressed in IPF where it localizes to aberrant epithelial regions [79] and functions as a Rho guanine nucleotide exchange factor regulating activation of RhoA [81], known for its involvement in profibrotic pathways.

## 7. Implicated Mechanisms

The precise pathogenesis of IPF is still not entirely understood, but the current knowledge on environmental and genetic risk factors strongly suggests epithelial injury-triggered reactivation of developmental pathways which, ultimately, leads to aberrant repair and regeneration resulting in drastic changes in lung structure and function. Therefore, in the following we will recapitulate these processes with a focus on what is known for the contributions of the bronchial and bronchiolar epithelium.

### 7.1. Types of Epithelial Injury

The airway epithelium represents the first line defence against inhaled particles, pathogens, and toxicants. Environmental and occupational triggers like cigarette smoke, wood dust, metal dust, pesticides, and herpesvirus infection are established risk factors for IPF [82,83]. Additionally, inhalation of traffic-related air pollutants has been linked to increased incidence of IPF [84]. Furthermore, gastroesophageal reflux (GER) is an overrepresented comorbidity of IPF, suggesting that microaspiration of stomach acids increases risk for IPF. Moreover, treatment of GER in IPF patients decelerates IPF disease progression and improves survival, indicating that GER also influences disease progression [82,83].

### 7.2. Epithelial Apoptosis

Apoptosis of alveolar epithelial cells is a well-established phenomenon in IPF and clearly reflected by the above discussed scRNA-Seq data showing a drastic decrease in normal alveolar type I and II cells in IPF relative to control lung tissue (Figure 3D). Immunofluorescent stainings of pro- and anti-apoptotic proteins in combination with terminal deoxynucleotide transferase-mediated deoxyuridine triphosphate-biotin nick end-labeling (TUNEL) stainings for DNA strand breaks have revealed that bronchiolar epithelial cells, hyperplastic epithelial cells and epithelial cells lining honeycomb cysts in the lungs of IPF patients show distinct signs of ongoing apoptosis [85,86,87,88]. While such cells in the past have often been referred to as “hyperplastic AT2 cells” [88], our recently gained more detailed understanding of the arising epithelial subpopulations in IPF, thanks to the above-described scRNA-Seq studies, strongly suggests that these cells also include epithelial cells of a bronchiolar origin like activated hyperplastic basal cells. Moreover, strengthening a potential role of apoptotic SCGBB1A1^+^ secretory cells in IPF, a recent report has demonstrated that ablation of programmed cell death 5 (*PDCD5*) expression in these secretory cells, but not in AT2 cells protects from experimental lung fibrosis [89].

### 7.3. Endoplasmic Reticulum (ER) Stress as Trigger for Epithelial Apoptosis

ER stress is a well-established trigger of alveolar epithelial apoptosis in IPF [85,90], but has received less attention for bronchial or bronchiolar epithelial cells. Many types of epithelial injury linked to an increased IPF risk, as, e.g., herpesvirus infection, cigarette smoke, and particulate matter, have been shown to cause ER stress and induce the unfolded protein response (UPR), also in cultured bronchial epithelial cells [90,91,92]. An elegant recent study has provided an intriguing link between the *MUC5B* promoter polymorphism (see Section 6) and ER stress in secretory airway epithelial cells. Chen et al. not only demonstrated that central components of the UPR induced *MUC5B* expression in secretory airway epithelial cells in pulmonary fibrosis, but also were able to show that this induction is dependent on sequences within the promoter variant rs35705950 region which harbours the IPF risk variant. Notably, in a luciferase reporter assay, the minor risk allele T alone increased expression of *MUC5B* by almost two-fold. This study provides another piece of evidence that ER stress and induction of the UPR in bronchiolar cells likely also contributes to expression of *MUC5B*, impaired mucociliary clearance, and the development of IPF [93].

### 7.4. Ageing and Epithelial Senescence

IPF predominates in the elderly and is characterized by increased senescence in many cell types, presumably because of replicative exhaustion and/or repetitive injuries to the epithelium [94]. It is by now well established that epithelial cells covering fibroblast foci are positive for senescence-associated β-galactosidase activity, nuclear p16 and p21 [95,96,97,98,99]. In agreement, recent scRNA-Seq-based studies have demonstrated that the above-described basaloid cells as well as hyperplastic basal cell population in bronchiolized regions express genes related to growth arrest and senescence [44,45,100]. This has also been observed for the transient population of Krt8^+^ ADI cells in mouse models of lung injury [54]. Collectively, these observations put forward an attractive hypothesis where a specific population of epithelial cells, normally committed to repair an injury of the lung mucosa followed by clearance, persists “locked in repair” in IPF [101]. Notably, senescent epithelial cells from fibrotic tissue have been shown to secrete proinflammatory and profibrotic molecules as components of their senescence-associated secretory phenotype (SASP) [97], suggesting that they may be a direct driver of disease pathogenesis.

### 7.5. Reactivation of Developmental Pathways

Reactivation of molecular signalling pathways such as the transforming growth factor-β (TGF-β), WNT, sonic hedgehog (SHH), and Notch pathways are critical players during the developmental stages of lung, remain largely inactive in the postnatal lung except for the maintenance of progenitor cell niches, but can become aberrantly reactivated during an injury repair response and then trigger chronic disease [102]. In the following, the induction and regulation of these developmental pathways during IPF pathogenesis is discussed with a focus in bronchial and bronchiolar epithelial cells.

#### 7.5.1. Transforming Growth Factor-β (TGF-β) Signalling

All three TGF-β isoforms (β1, β2, β3), their receptors TGF-β receptors (TGFBR) I, II, and III, and their signalling mediators SMAD-2, -3, -4, -5, -6 and -7 are involved in embryonic lung development where they regulate branching morphogenesis and alveolarization [102]. TGF-β ligands act by binding to their cognate receptors on target cells, where they trigger intracellular signalling pathways including the canonical SMAD-mediated pathway but also non-canonical signalling pathways [103].

TGF-β is synthesized as an inactive precursor homodimer with N-terminal prodomains, which, after cleavage by the intracellular protease furin, remain non-covalently bound to the TGF-β homodimer as latency-associated peptide (LAP), collectively forming the small latent complex (SLC). Only if this complex is bound to the latent TGF-β-binding protein (LTBP), it will be secreted to the extracellular matrix as a complex called large latent complex (LLC) [104]. Hence, TGF-β is always secreted in a latent form and requires activation in situ by additional triggers.

Out of the three isoforms, TGF-β1 plays a well-recognized central role in IPF pathogenesis [105,106,107]. Activation of latent TGF-β1 implies the release of active TGF-β1 ligands from the ECM by proteolysis or deformation of their LAP portion. Many potential mechanisms have been observed in vitro, but for many the physiological relevance remains unclear. In vivo activation has been clearly shown for several αv integrins in the context of fibrosis, e.g., avβ1, avβ3, avβ5, and avβ6 [108]. Even though the underlying mechanisms are not fully understood, it appears that cells carrying these integrins can exert a pulling force on the LLC which “unwraps” the LAP and releases active TGF-β1 from the ECM. Other reasonably well-established activators are thrombospondin-1 (TSP1), pregnancy specific glycoproteins, and tenascin X. Additionally, activation by unspecific physico- or biochemical factors like low pH and reactive oxygen species has been described, which may also be physiologically relevant. Finally, proteolytic activation has been described for a variety of proteases, including, e.g., several matrix metalloproteinases (MMPs), calpain, plasmin, kallikrein, and cathepsin D. Interestingly, while deficiency of integrin subunits like αv, β6, and β8 in mice phenocopies the TGF-β1 knockout mouse, this has not been observed for any protease-deficient mouse so far, indicating considerable redundancy in proteolytic activation of TGF-β1 in vivo [108,109,110].

Bronchial epithelial cells potentially may contribute to TGF-β1-mediated mechanisms in IPF by at least three mechanisms. First, bronchial and bronchiolar epithelial cells express TGF-β1 [111,112], implying that the underlying ECM likely harbours latent TGF-β1. Second, bronchial epithelial express many of the suggested activating factors in fibrosis: Airway epithelial cells express both αvβ6 and αvβ8 integrin heterodimers, and expression of αvβ6 is dramatically increased after injury [113]. Notably, the *ITGAV* transcript for the αv integrin monomer is clearly enriched in aberrant basaloid cells relative to all other healthy epithelial cell types (Figure 5A). Second, airway and aberrant basaloid epithelial cells also have been shown to express activators of latent TGF-β1 in IPF, including MMP-8 [114], MMP-3, MMP-13, MMP14, calpain, and cathepsin D in IPF [44,45] (ipfcellatlas.com), all representing proteases previously proposed to activate latent TGF-β1 [108]. ScRNA-Seq data also demonstrates expression of the thrombospondin 1 precursor by bronchial epithelial cells [44,45] (ipfcellatlas.com). Third, bronchial epithelial cells themselves are reactive to TGF-β1 and have been shown to undergo partial epithelial-to-mesenchymal transition (pEMT) in response to TGF-β1 [115,116]. Whether EMT contributes to the myofibroblast population in IPF is controversially discussed, as conflicting results have been reported in in vivo models of pulmonary fibrosis—so far neither lineage-tracing experiments nor scRNA-Seq data have provided unambiguous evidence for a complete EMT as a source for myofibroblasts in the lung [117,118]. However, the resulting cell phenotype after pEMT is partly reminiscent of the aberrant basal-like cell phenotype observed in IPF—following pEMT, human bronchiolar epithelial cells lose epithelial morphology and polarity and upregulate mesenchymal markers like type I collagen and fibronectin. On the other hand, downregulation of expression of typical epithelial markers such as E-cadherin and upregulation of vimentin is not evident in the scRNA-Seq data sets published so far [44,45] (ipfcellatlas.com). These discrepancies may reflect the crosstalk between variously activated profibrotic pathways and the complex cellular and ECM environment in end-stage IPF, parameters frequently not considered in studies of EMT. Clearly, further work is warranted to elucidate the role of TGF-β1 in the emergence of aberrant basaloid cells, and how this process relates to pEMT.

#### 7.5.2. WNT Signalling Pathway

Wingless/integrase-1 (WNT) signalling pathways are fundamentally important for tissue morphogenesis including all stages of lung development [119]. The WNT ligand family comprises 19 human members which are characterized by strictly controlled spatiotemporal expression in various organs during development and tissue homeostasis and associated with a constantly growing number of human diseases by upregulation, genetic polymorphisms and mutations [120]. It is well-established that the WNT signalling pathway is reactivated in IPF [119,121] and expression of WNT ligands (WNT1, WNT3a), intracellular downstream inducers (β-catenin, GSK-3β), as well as extracellular inhibitors of canonical WNT signalling (Dickkopf proteins DKK1, DKK4 and the interacting transmembrane receptor Kremen 1) has been demonstrated in bronchial and bronchiolar epithelium in IPF [122,123]. Studies in various models of lung injury have put forward WNT signalling as a critical component for stem cell maintenance, lung regeneration, and repair [119]. WNT signalling is activated during repair after proximal lung injury and dynamically regulates submucosal gland progenitor maintenance, proliferation, and differentiation to other airway epithelial cell types [124,125,126,127,128]. Furthermore, in mice, expression of Wnt7b by basal cells in the proximal airways generates their own stem cell niche via induction of fibroblast growth factor 10 (Fgf10) in adjacent smooth muscle cells [129]. Airway injury induces Wnt7b in the more distal airways, generating new Fgf10-expressing mesenchymal cells and allowing for recruitment of basal cells and/or differentiation of lineage-negative progenitors into the basal progenitor cell lineage [129,130]. Collectively, these studies imply an important role of WNT signalling in aberrant bronchial and bronchiolar repair in IPF.

#### 7.5.3. Sonic Hedgehog Signalling (SHH) Pathway

During lung development, sonic hedgehog (SHH) is expressed in the respiratory epithelium in a gradient with higher levels in the branching tips, presumably providing polarization during branching morphogenesis in the embryonic and pseudoglandular stage. Furthermore, SHH is essential for the coordination of epithelial-mesenchymal compartment growth, also during the alveolarization phase [131,132]. Bolaños et al. systematically assessed expression of SHH signalling pathway components in control lung tissue and IPF and found that expression of all SHH signalling components was induced or drastically increased in IPF. They observed expression of the ligand SHH exclusively in bronchial, bronchiolar, and alveolar epithelial cells, but expression of the receptors transmembrane receptor Patched-1 and the G-protein coupled receptor Smoothened mainly in fibroblasts and inflammatory cells. While the SHH signalling transcription factor glioma-associated oncogene homolog (*GLI*) *1* was expressed ubiquitously, including in fibroblasts, nuclear GLI2 was confined to distal epithelial cells [133]. Furthermore, the authors could show that recombinant SHH increased proliferation, expression of ECM components, and migration of primary human lung fibroblasts and at the same time inhibited fibroblast apoptosis [133]. These results indicate that SHH generated by distal, bronchiolar and alveolar, epithelial cells activates fibroblasts, which indicates an important profibrotic contribution of epithelial-derived SHH in IPF pathogenesis. Interestingly, a more recent study provided evidence that a profibrotic feed-forward mechanism may exist in this context: Gli^+^ mesenchymal stromal cells promote differentiation of airway progenitors into aberrant metaplastic Krt5^+^ basal cells by antagonizing activation of the bone morphogenetic protein (BMP) pathway [134]. Overall, this suggests that upregulation of epithelial SHH may be an early event in IPF pathogenesis and trigger reciprocal epithelial-mesenchymal interactions that propagate lung fibrogenesis.

#### 7.5.4. Notch Signalling Pathway

In lung development, Notch signalling determines ciliated versus secretory cell fate in conducting airways [135,136]. Following bleomycin injury or influenza infection in mice, Notch signalling has been shown to activate proliferation and migration of a KRT5^+^ progenitor cell lineage in the context of repair after injury while blockade of Notch signalling induced an alveolar cell type faith. Importantly, active Notch signalling was detected in IPF honeycomb cysts [130], indicating a role for Notch signalling in aberrant epithelial repair and honeycomb cyst formation. Interestingly, overexpression of Notch can also induce EMT [137]; so, Notch signalling may not only promote aberrant cyst formation, but also contribute to the emergence of the above- described aberrant basaloid cells. In mice, Dlk1-mediated temporal regulation of Notch signalling is required for differentiation of AT2 to AT1 cells during repair [138]. Interestingly, deletion of Dlk1 in AT2 cells led to the accumulation of an intermediate cell population. We may speculate that a similar Notch-dependent mechanism might drive the appearance of aberrant basaloid cells in IPF.

In summary, bronchial and bronchiolar epithelial cells including airway-cell derived disease-specific lineages contribute to the reactivation of developmental pathways in IPF, including central pathways like the TGF-β1, WNT, SHH, and Notch signalling pathways. The collective evidence clearly demonstrates that, via autocrine and paracrine mechanisms, conducting airway epithelial-derived factors induce and modulate developmental programmes in IPF and drive major pathological outcomes in this disease like excessive ECM deposition and honeycomb cyst formation.

### 7.6. Epigenetic Mechanisms

Epigenetics traditionally comprises DNA methylation and histone modification, molecular alterations in chromatin which serve as marks for transcriptional activation or repression without affecting the DNA sequence per se. Epigenetic regulation mechanisms are typically persistent, can be inherited, and have the potential to translate environmental exposures into regulation of gene transcription at the level of chromatin structure [139,140]. This applies particularly to the airway mucosa, which represents a direct interface between environment and human body [141,142]. As IPF development seems to be orchestrated by genetic predisposition and environmental risk factors, epigenetic mechanisms may provide important mechanistic links and novel targets for therapy. Indeed, a number of studies have established that epigenetic signatures are changed in IPF, including DNA methylation and expression of DNA methyl transferases [143,144] as well as single histone modification marks [140] and expression of histone modifying enzymes [145]. To the best of our knowledge, genome-wide histone modification studies in IPF are lacking to date.

Our knowledge on epigenetic marks in IPF and their cell type-specific contribution to disease pathogenesis and progression is still very limited. However, it is well-known that IPF risk factors like cigarette smoke or particulate matter, for instance, induce epigenetic alterations in bronchial epithelial cells [146,147,148], indicating that such changes may be frequent in IPF. Furthermore, increased expression and activity of histone deacetylases in IPF has been localized to myofibroblasts, but also to aberrant basal cells in IPF [145]. Clearly, the role of epigenetic changes in airway epithelial cells requires more attention and detailed mechanistic studies, and such investigations may ultimately provide interesting novel therapeutic intervention opportunities for early therapy.

### 7.7. Non-Coding RNAs

Non-coding RNA (ncRNA), i.e., RNA which is not translated to proteins, constitutes approximately 98% of the total transcribed RNA in humans [149]. NcRNAs include housekeeping RNAs, such as ribosomal, spliceosomal, or transfer RNA, expression of which is constitutive, but also regulatory RNAs, such as long noncoding RNAs (lncRNA) or microRNAs (miRNA), which are expressed in a cell type- and tissue-specific manner and often altered in disease. LncRNA molecules are arbitrarily defined as >200 nucleotides in length and can regulate gene expression by transcriptional interference, chromatin remodelling, promoter inactivation, activation and transport of accessory and transcription factors, epigenetic silencing, and as precursors for small interfering RNAs [150,151]. In contrast, miRNAs are short, approximately 22 nucleotides long, RNA molecules which suppress protein translation by non-perfect complementary binding to regions in the 3′UTR of their target mRNAs.

Even though our knowledge on function and regulation of lncRNAs in general is still very limited, several studies support the concept that lncRNAs contribute to profibrotic cellular mechanisms in IPF [152,153]. While some studies in this context focussed on the function of specific lncRNAs in lung fibroblasts [154], other recent reports highlight altered lncRNA expression and function in bronchial epithelial cells. For instance, increased expression of lncRNA *MEG3* was observed in atypical KRT5^+^ p63^+^ basal cells in IPF relative to normal donor lung tissue. In vitro studies showed that *MEG3* induced basal cell gene transcription (*KRT14, TP63*) in bronchial cell lines, but also fundamental events of EMT, including increased cellular migration and downregulation of *CDH1* (E-cadherin) [155]. *MEG3* may thus cause or at least contribute to the emergence of the aberrant basal-like cell populations in IPF described above (see Section 4). In contrast, loss of the terminal differentiation-induced lncRNA (TINCR), a lncRNA normally expressed in the bronchial epithelium, but decreased in IPF, has been described to, among others, induce basal cell markers and ECM genes [156,157], reminiscent of gene expression signatures of aberrant basal and basaloid cells in IPF [42,44,45]. Studies in mouse models of lung fibrosis and primary human cells have proposed additional lncRNAs as regulators of EMT in bronchial epithelial cells, but localization in the IPF lung has, to the best of our knowledge, not yet been demonstrated. These include lncRNAs uc.77 and 2700086A05Rik [158] and lncRNA H19 [159]. Collectively, these studies support the concept of bronchial epithelial cell-specific lncRNA expression as an emerging driver in IPF pathogenesis.

To date, few studies have addressed the function of airway epithelial miRNAs in IPF pathogenesis. A pioneering study has globally assessed expression of miRNAs in bronchoscopy-assisted bronchial brushes from fibrotic airways of bronchiolitis obliterans syndrome (BOS) and found that miR-323a-3p was drastically downregulated (>18-fold) in airways of BOS patients relative to control lung transplant patients. The authors also examined miR-323a-3p expression in isolated AT2 cells from IPF lung explants and from fibrotic mouse lungs after bleomycin injury and observed significant downregulation, indicating general downregulation in lung epithelium during fibrogenesis [160]. Furthermore, miR-323a-3p mimics and miR-323a-3p antagomirs suppressed and exacerbated lung fibrogenesis, respectively, in the bleomycin mouse model. In vitro studies suggested that miR-323a-3p directly targets central mediators of TGF-α and TGF-β signalling as well as caspase 3, thereby attenuating key profibrotic mechanisms and epithelial cell apoptosis [160]. Given that miRNA therapeutics are coming of age and, in the case of the lung, can be easily delivered to the epithelium by inhalation, more such studies are warranted to identify further epithelial-specific miRNA-based profibrotic mechanisms.

## 8. Summary, Conclusions, and Emerging Questions

The last decade has transformed our understanding of IPF pathogenesis and set forth multiple evidence that strongly argues for a critical role of conducting airway epithelial cell populations in IPF aetiology and disease development (summarized in Figure 7). The discovery of the *MUC5B* promoter polymorphism as the strongest causative factor for IPF onset drew attention from the alveolar department to bronchial and bronchiolar cell contributions to lung fibrogenesis. IPF airways are drastically distorted, and alveolar areas are repopulated by airway-like epithelial cells in a process termed bronchiolization. In agreement, several recent scRNA-Seq analyses of IPF lungs have consistently revealed drastic alterations in epithelial subpopulations including the replacement of alveolar epithelial cells by various airway-like cells that are either directly distal airway-derived or the result of alveolar epithelial cell transdifferentiation or a combination of both. Emerging new evidence suggests that specific mesenchymal niche environments in the IPF patient may promote plasticity of the alveolar epithelium that leads to full transdifferentiation towards airway-like states [38]. Another line of evidence shows that persistent alveolar repair generates intermediate cells, which display features of senescence and p53 activation. In mice, inducing senescence in AT2 cells and thereby shifting them to a state that resembles injury-induced alveolar differentiation intermediates [54,161] and the aberrant basaloid cells [42,44,45] leads to progressive pulmonary fibrosis as seen in IPF patients [162]. Future work needs to leverage histopathological disease grade staging to further clarify the cellular origins of these intermediate cell populations and the natural evolution of epithelial metaplasia and bronchiolization in IPF disease progression.

Critical airway functions like mucociliary clearance and epithelial barrier integrity are also affected in IPF. Genetic risk factors beyond the *MUC5B* promoter polymorphism, in particular the *DSP* and *FAM13A* risk SNPs, argue for airway epithelial cells as central culprits in disease onset. Finally, evidence is accumulating that bronchial epithelial cells directly trigger central profibrotic mechanisms like the reactivation of multiple developmental programmes in an aberrant injury response.

The balance between epithelial proliferation, trans-differentiation, apoptosis and cellular senescence is drastically disturbed in IPF airway epithelial cells. Impaired mucociliary clearance may be a key disease-initiating feature in this context. However, we still understand very little about the mechanisms that trigger the balance to tip from normal alveolar repair towards this aberrant, airway epithelial cell-driven repair process leading to the emergence of epithelial metaplasia and aberrant basaloid cells in the lung periphery. Similarly, the sequence of events that ultimately lead to IPF development remains ill-defined. For instance, is bronchiolization an epiphenomenon and characteristic of end-stage disease, or may pEMT of airway epithelial cells actually precede activation of fibroblasts? What are key mechanisms that can be safely and effectively employed to target profibrotic epithelial-mesenchymal cross-talk and regenerate normal stem cell niches? In particular epigenetic mechanisms, the role of epithelial non-coding RNAs, how these affect profibrotic and disease-perpetuating mechanisms, and whether they can be targeted for therapy remains a largely unexplored area. Additionally, the contributions of immune cells to the described processes remain little understood. Evidently, more mechanistic studies are needed to decipher these processes in molecular detail. It is becoming increasingly clear that, for this aim, we need to develop novel animal lung fibrosis models, which recapitulate impaired mucociliary function and environmental exposure. The above-described mouse model derived by conditional deletion of NEDD4-2 from airway epithelial cells represents a great opportunity to study in more detail the mechanisms that trigger fibrosis as a result of impaired mucociliary clearance. The good news about airway epithelial cells as emerging central culprits in IPF pathogenesis is that, finally, targeting airway epithelial cells is a more straightforward task than targeting fibroblasts, because, given that fibrotic areas are ventilated, the inhalatory route would deliver the drug directly and specifically onto the aberrant epithelium.

## Figures and Tables

**Figure 1 cells-11-01050-f001:**
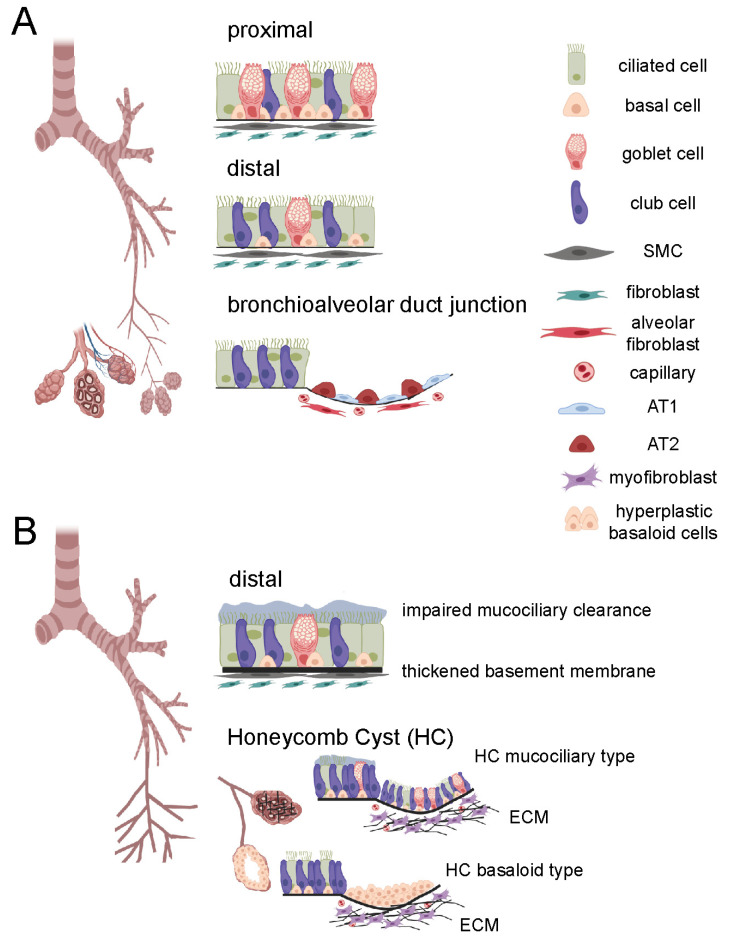
Schematic overview of airways in healthy lung and idiopathic pulmonary fibrosis (IPF). (**A**) Airways in the healthy lung, depicting normal cell type distribution in the proximal and distal airways as well as in the bronchioalveolar duct junction. (**B**) Airways in the IPF lung, depicting dilated bronchioles, impaired mucociliary clearance and the thickened basement membrane in the distal airways, two types of honeycomb cysts (HC, mucociliary, basaloid), and accumulation of extracellular matrix (ECM) in the alveolar region. AT1, alveolar cell type 1; AT2, alveolar cell type II; ECM, extracellular matrix; SMC, smooth muscle cell. Figure was created with biorender.com.

**Figure 2 cells-11-01050-f002:**
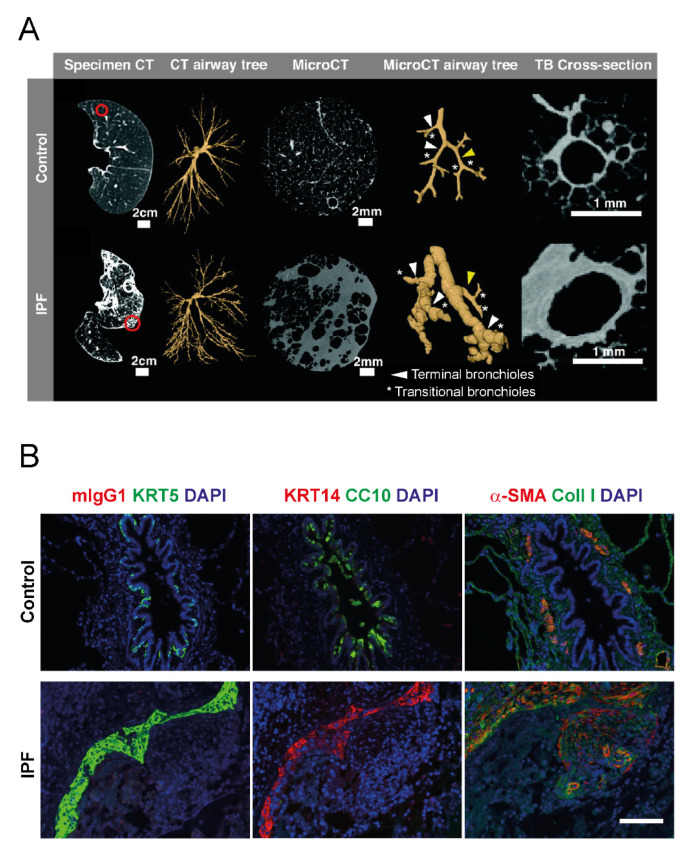
Airway epithelial abnormalities in IPF. (**A**) Comparison of airway features in control and IPF lungs as monitored by computed tomography (CT, adapted from Ikezoe et al. [12] with permission of the American Thoracic Society). Computed tomography (CT) scans from lungs of a control subject (upper row) and a case of IPF (lower row). The panels show from left to right: (1) Axial midslice multidetector computed tomography (MDCT) scans indicating where a random tissue sample was obtained for microCT (red circles); (2) reconstructed airway tree for the same scan from the lateral perspective; (3) midslice microCT scans of the tissue sample circled in red; (4) small airway tree segmentations obtained from the microCT scans visualized in three dimensions, identifying terminal bronchioles (TB, white arrowheads) and transitional bronchioles (asterisks); (5) representative cross-sectional image of the terminal bronchiole (TB) highlighted by the yellow arrowhead. This figure panel is adapted from Ikezoe et al. [12] with permission of the American Thoracic Society. Copyright © 2022 American Thoracic Society. All rights reserved. The American Journal of Respiratory and Critical Care Medicine is an official journal of the American Thoracic Society. Readers are encouraged to read the entire article for the correct context at https://www.atsjournals.org/doi/10.1164/rccm.202103-0585OC (last accessed 8 March 2022). The authors, editors, and The American Thoracic Society are not responsible for errors or omissions in adaptations. (**B**) Immunofluorescent stainings of serial lung sections of a representative control subject (upper row) and a case of IPF (lower row) with mouse isotype control antibody (mIgG1) and antibodies directed towards keratin 5 (KRT5), keratin 14 (KRT14), club cell-specific protein 10 (CC10), α-smooth muscle actin (α-SMA) as a marker for smooth muscle cells and myofibroblasts, and type I collagen (Coll I). Scale bar 100 µm.

**Figure 3 cells-11-01050-f003:**
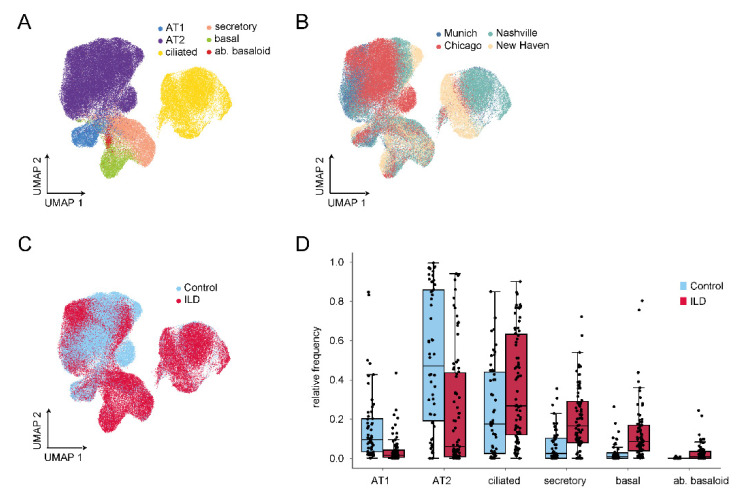
Single cell RNA-Sequencing has revealed drastic changes in epithelial cell populations in ILD. (**A**) Uniform Manifold Approximation and Projection (UMAP)-based dimension reduction of single cell transcriptomic data to delineate epithelial cell types, labelled by cell type. (**B**) Same UMAP visualization labelled by ILD cohort. Data used for visualization was derived from in total four datasets [42,43,44,45] of control and interstitial lung disease (ILD) samples: New Haven [45], Nashville [44], Chicago [42], and Munich [43]. (**C**) Same UMAP visualization labelled by disease. (**D**) Relative frequencies of epithelial cell populations demonstrate a consistent increase in conducting airway cell populations in ILD at the expense of alveolar type 1 (AT1) and 2 (AT2) cells. ab., aberrant.

**Figure 4 cells-11-01050-f004:**
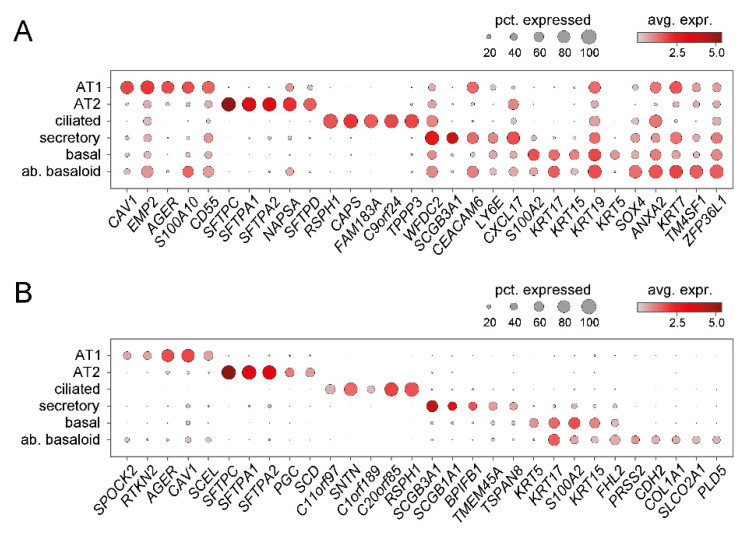
Cell type-specific markers for epithelial cell populations in ILD derived from scRNA-Seq data. Using the data set described in Figure 3, the top 5 specific markers for the described epithelial populations are plotted, (**A**) ranked by adjusted *p*-value or (**B**) ranked by log fold changes of relevant cell type vs. all other epithelial cell types. pct., percentage; avg. expr., average expression; ab., aberrant.

**Figure 5 cells-11-01050-f005:**
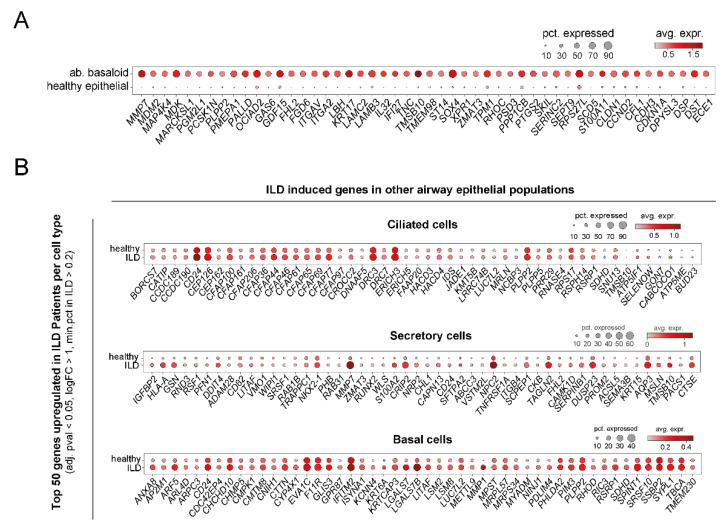
Epithelial cell populations show distinct expression changes in ILD. Using the data set described in Figure 3, differential gene expression analysis was performed with diffxpy (https://github.com/theislab/diffxpy, last accessed 22 December 2021) while accounting for number of transcripts per cell and patient cohort. The top 50 deregulated genes in specific subpopulations of epithelial cells are given, ranked by log2 fold change. (**A**) Top 50 genes induced in aberrant basaloid cells relative to gene expression of all other healthy epithelial cell types. (**B**) Top 50 genes increased in ILD in other airway epithelial cell populations. pct., percentage; avg. expr., average expression; ab., aberrant.

**Figure 6 cells-11-01050-f006:**
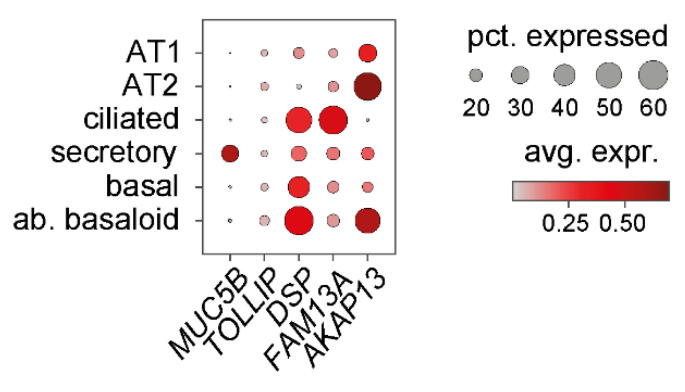
Expression of selected risk factor genes in epithelial cell populations. Using the data set described in Figure 3, expression of selected genes harbouring IPF risk-associated SNPs is given. Selection was based on previous reports on their expression in airway epithelium (see text for more details). pct., percentage; avg. expr., average expression; ab., aberrant.

**Figure 7 cells-11-01050-f007:**
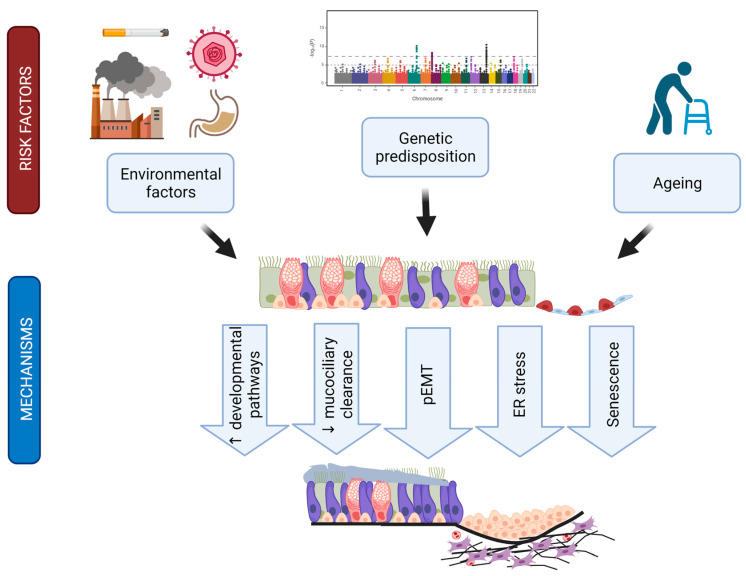
Hypothetical contributions of the airway epithelium to IPF pathogenesis. Summarizing scheme linking established environmental and genetic risk factors via the bronchial and bronchiolar epithelium to IPF-specific disease mechanisms and outcomes like bronchiolization and interstitial scarring. Figure was created with biorender.com.

## Data Availability

Count tables of the Munich single-cell cohort as well as custom preprocessing code can be accessed at https://github.com/theislab/2020_Mayr (last accessed 22 December 2021). Raw count tables for additional cohorts were retrieved from the Gene Expression Omnibus database by the accession numbers as provided in the original publications (Chicago cohort GSE122960; Nashville cohort GSE135893; New Haven cohort GSE136831).

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
