# Peer review of "Emerging Roles of Airway Epithelial Cells in Idiopathic Pulmonary Fibrosis"

_cells, 2022, doi:10.3390/cells11061050_

Round 1

Reviewer 1 Report

Emerging Roles of Airway Epithelial Cells in Idiopathic Pulmonary Fibrosis

This is a review article by Chakraborty et al described the involvement of airway epithelial cells in lung fibrosis. The authors’ work is comprehensive and complete. The roles of airway epithelial cells have gained significant interest recently, mostly described in various transcriptomic studies of IPF and ILD explant lungs that were integrated in clinical characteristics. A lack of reliable animal model to implicate the role of airway epithelia is a major challenge. Ultimately, at the current stage, pathological functions of these cells in lung fibrosis remain elusive. And therefore, schematic figure 6 can be quite presumptuous. 

Comments
1. In this review, a bit of the underlying mechanisms of pulmonary fibrosis were derived from AT2 cells. To make the review more concise and not too lengthy, focusing on only evidence related to airway epithelial cells would help.
2. Four of 6 figures are transcriptomic data, if space permits, clinical illustration such as imaging or histology will provide additional emphasis on the role of airway epithelial cells.

Author Response

Comment 1: Ultimately, at the current stage, pathological functions of these cells in lung fibrosis remain elusive. And therefore, schematic figure 6 (edit: now figure 7) can be quite presumptuous. 

Answer 1: We thank the reviewer for this comment and fully agree. We really did not want this figure to come across as such a strong statement, or to make believe that bronchial epithelial cells are the ultimate and only culprit in IPF. It was supposed to represent more like a hypothetical statement, highlighting the ways how airway epithelial cells max contribute to the overall IPF disease pathogenesis. This is now clarified in the figure legend of figure 7, where we have replaced “The airway epithelium takes centre stage in IPF” with a more carefully phrased statement: “Hypothetical contributions of the airway epithelium to IPF pathogenesis.”

Comment 2: In this review, a bit of the underlying mechanisms of pulmonary fibrosis were derived from AT2 cells. To make the review more concise and not too lengthy, focusing on only evidence related to airway epithelial cells would help.

Answer 2: Thank you for this comment, we agree and have shortened the corresponding paragraph on page 4 (from line 154). We also considered shortening the first paragraph in the scRNA-Seq section which is mostly referring to Epcam+/HTII-280+ cells (i.e. AT2 cells), but finally left it in the review because of reviewer 2’s suggestion of adding Wasnick et al, Cells, 2022 (PMID: 35053350) which takes a similar initial approach, focusing on HTII-280+ cells.

Comment 3: Four of six figures are transcriptomic data, if space permits, clinical illustration such as imaging or histology will provide additional emphasis on the role of airway epithelial cells.

Answer 3: Thank you for this excellent suggestion. We have included such a figure now (new figure 2) showing both computed tomography-based assessments as well as histological findings from our lab.

Reviewer 2 Report

With real interest, I read the manuscript cells-1602166.

It is a very nice review combining bride knowledge of the Authors with unbiased conclusions made based on the statistical and bioinformatic analyses of the publicly available datasets, including those created by the Authors themselves.

I have only some minor and/or facultative comments:

  1. In the very same special issue to which the Authors submitted the current work, a very interesting article (PMID: 35053350) has been published that should be included.
  2. Please, be careful with the details, e.g. “ATII” vs. “AT2”. Either or/unify, please.
  3. For the figures 2-5, please, provide some more methodological details. Even if the used software is obvious, please, report it where not done.
  4. The Authors are very careful with writing genes in italics. Still, one more check would be good (e.g. line 267).
  5. Besides, names of the genes should be written in italics also in the figures (Figures 3-5).
  6. I have a feeling that the title of the chapters are written with different font than the text.
  7. Lines 528-531. Please, add some newer papers defining the basics of epigenetic mechanisms (PMID: 32973742, 33668787).

Author Response

Comment 1: In the very same special issue to which the Authors submitted the current work, a very interesting article (PMID: 35053350) has been published that should be included.

Answer 1: First, of all, we are grateful to the reviewer for this supportive assessment of our review. We agree that this reference should be included and have done so on page 7, line 241 (version with tracked changes).

Comment 2: Please, be careful with the details, e.g. “ATII” vs. “AT2”. Either or/unify, please.

Answer 2: Thank you for this observant comment – we have consistently used AT2 now in the revised version.

Comment 3: For the figures 2-5, please, provide some more methodological details. Even if the used software is obvious, please, report it where not done.

Answer 3: This is a very valid point, thank you. We have added the requested information in the text where we introduce the integration of the four data sets (page 7. lines 264 onward in the tracked changes version)

Comment 4: The Authors are very careful with writing genes in italics. Still, one more check would be good (e.g. line 267). Besides, names of the genes should be written in italics also in the figures (Figures 3-5).

Answer 4: Thank you for bringing this to our attention. We have carefully revised the manuscript accordingly, italicized gene and transcript symbols where applicable, including in Figures 4-6 (previous figures 3-5).

Comment 5: I have a feeling that the title of the chapters are written with different font than the text.

Answer 5: As we used the Cells template now for the revised version, this is not applicable anymore.

Comment 6: Lines 528-531. Please, add some newer papers defining the basics of epigenetic mechanisms (PMID: 32973742, 33668787).

Answer 6: Thank you for this excellent suggestion – we have included the mentioned references in that section, highlighting the prominent role of the airway epithelium as an interface between environment and human body (see page 18, lines 749-750).